# Pharmacological Modulation of Steroid Activity in Hormone-Dependent Breast and Prostate Cancers: Effect of Some Plant Extract Derivatives

**DOI:** 10.3390/ijms21103690

**Published:** 2020-05-23

**Authors:** Bagora Bayala, Abdou Azaque Zoure, Silvère Baron, Cyrille de Joussineau, Jacques Simpore, Jean-Marc A. Lobaccaro

**Affiliations:** 1Laboratoire de Biologie Moléculaire et de Génétique (LABIOGENE), Université Joseph KI-ZERBO, 03 BP 7021 Ouagadougou 03, Burkina Faso; zabdouazaque@yahoo.fr (A.A.Z.); jacques.simpore@yahoo.fr (J.S.); 2Centre de Recherche Biomoléculaire Pietro Annigoni (CERBA), 01 BP 216 Ouagadougou 01, Burkina Faso; 3Université Clermont Auvergne, CNRS, Inserm, GReD, F-63000 Clermont-Ferrand, France; Silvere.baron@uca.fr (S.B.); cyrille.de_joussineau@uca.fr (C.d.J.); 4Université Norbert ZONGO, BP 376 Koudougou, Burkina Faso; 5Département Biomédical et Santé Publique, Institut de Recherche en Sciences de la Santé (IRSS/CNRST), 03 BP 7192 Ouagadougou 03, Burkina Faso

**Keywords:** nuclear receptors, hormone-dependent tumors, prostate and breast cancers, plant extract derivatives, steroids

## Abstract

The great majority of breast and prostate tumors are hormone-dependent cancers; hence, estrogens and androgens can, respectively, drive their developments, making it possible to use pharmacological therapies in their hormone-dependent phases by targeting the levels of steroid or modulating their physiological activity through their respective nuclear receptors when the tumors relapse. Unfortunately, at some stage, both breast and prostate cancers become resistant to pharmacological treatments that aim to block their receptors, estrogen (ER) or androgen (AR) receptors, respectively. So far, antiestrogens and antiandrogens used in clinics have been designed based on their structural analogies with natural hormones, 17-β estradiol and dihydrotestosterone. Plants are a potential source of drug discovery and the development of new pharmacological compounds. The aim of this review article is to highlight the recent advances in the pharmacological modulation of androgen or estrogen levels, and their activity through their cognate nuclear receptors in prostate or breast cancer and the effects of some plants extracts.

## 1. Introduction

The second leading cause of mortality worldwide [1], cancer is a complex situation. This is based in part on the extreme heterogeneity of the genetic causes, the levels of the secreted growth factors and circulating hormones, and the interactions between the tumor cells and the surrounding microenvironment [2]. Extensive research over the past 25 years in breast (BCa) and prostate (PCa) cancers have deciphered the molecular mechanisms driven by steroid receptors and elucidated the interplay between genomic and non-genomic activities of these steroid receptors. Altogether, these mechanisms pilot specific gene expression programs that contribute to the tissue homeostasis and the disequilibrium in such a subtle balance that could induce endocrine therapy resistance and cancer progression [3]. The way normal cells transform into cancer cells and how they maintain their malignant state and phenotypes have been associated with genetic and epigenetic deregulations [4]. Interestingly, these alterations are constantly evolving as tumor cells face changing selective pressures such as drug treatments. This is why the discovery and development of new therapeutics is a real challenge for bypassing tumor escapes.

Historically, plants, animals, fungi, and microorganisms have been extensively used as a source of biologically active compounds. Hence, despite the development of chemistry and the introduction of synthetic chemotherapeutics, a substantial part of new pharmaceuticals still remains of a natural origin [5]. Indeed, natural compounds exhibit great diversity in the chemical structures and, thus can act on diverse mechanisms of action through different molecular targets, such as modulating the levels of circulating hormones, blocking or activating nuclear or membrane receptors, or targeting nucleic acids as described for antimicrobial compounds [5]. Globally, evolution has provided numerous suitable candidates for anti-cancer drug discovery due to their pleiotropic activity on target molecules [6]. Among the various types of tumors, hormone-dependent cancers are an exquisite example of what could be done in order to regulate the activity of hormones.

Enzymatically derived from cholesterol (Figure 1), sex steroids are mainly synthetized in the gonads, even though adrenal glands could also produce non-active precursors that may eventually be transformed into active androgens or estrogens [7]. Basically, testosterone is transformed into 5α-dihydrotestosterone (DHT) by the 5α-reductase before acting through AR on prostate growth, while aromatase (present in breast and adipose tissues) acts on testosterone to produce 17β-estradiol (17βE2) acting through ERα. It was also noted that ∆4-androstenedione could be considered as an important adrenal precursor as it could be transformed into testosterone or estrone by 17β hydroxysteroid dehydrogenase (17βHSD).

Schematically, several pathways could be pharmacologically targeted to suppress, or at least reduce, the effects of the steroid hormones in BCa and PCa: (i) the modulation of the enzymes synthetizing testosterone, 17βE2 and DHT namely 17βHSD, aromatase, and 5α-reductase; (ii) specifically preventing the binding of 17βE2 or DHT to their cognate nuclear receptor, respectively, estrogen (ER; NR3A1/2) and androgen (AR; NR3CA) receptors, by the use of specific estrogen (SERMs) or specific androgen (SARMs) receptor modulators; (iii) the specific degradation of ER or AR by antagonists blocking the effect of the natural hormones as well as decreasing the quantity of the receptors, like specific ER degraders or down-regulators (SERDs) or specific AR degraders (SARDs) (Figure 2).

This review aims to focus on the pathways that could be targeted to modify the activity of steroids. Some of the natural compounds described as promising therapeutics to treat BCa or PCa will be indicated, as well as some of their pharmacological activity as modulators of AR and ERα levels and modifiers of their hormonal response.

## 2. PCa and BCa, Two Hormone-Dependent Cancers

Prostate cancer (PCa) has a long natural history from the diagnosis to the death caused by cancer progression [8]. While androgens have been described to be necessary for the development and maintenance of the prostate gland; they are also responsible for the development of the tumor [9]. Schematically, a prostate tumor is composed of multiple epithelial cell types, inter-mingled with various fibroblasts, neuroendocrine cells, endothelial cells, macrophages and lymphocytes, all of them interacting to influence treatment responses in a patient-specific manner [10]. Androgens and their receptor (AR), thus play a key role in the development of prostate tissue by guiding cytodifferentiation and homeostasis of normal or tumor luminal epithelial cells. Various risk factors may lead to prostate carcinogenesis, including infectious agents [11], contamination by heavy metals [12], alcohol, and tobacco consumption [13]. Based on its heterogeneity, the management of the patients diagnosed with a PCa depends on the stage of the tumor, the age of diagnosis, and the expected quality of life (for an extensive review see [14]). Hence in local PCa, which represents about 80% of diagnosed PCa disease [15], the European Association of Urology guidelines propose a radical prostatectomy to patients with low to high-risk PCa since they have a life expectancy >10 years. Radiation therapy is a suitable option for low-risk PCa and should be used in combination with androgen deprivation therapy (ADT) for intermediate/high-risk localized and locally advanced PCa. In advanced and metastatic PCa (about 5% of the tumors), the median survival, even heterogeneous, is around 42 months. The first line standard approach is ADT [16]. Besides blocking androgen effects that will be developed above, treatment with gonadotropin releasing hormone (GnRH) analogs results in a significant decrease of luteinizing (LH)/follicle-stimulating (FSH) hormone secretions and then testosterone production. Together with ADT, chemotherapy may be performed as well [14]. Despite ADT, most patients experience tumor growth recovery within a median of 18 to 24 months and progress to a lethal stage called castration-resistant PCa (CRPC). The emergence of this aggressive form of PCa is diagnosed when blood PSA increases despite low serum testosterone. CRPC is followed by a further progression of the disease with the appearance of new symptoms and bone or soft tissue lesions [17].

As for prostate tropism and androgens, the growth of BCa is mainly related to in situ levels of estrogens and the stimulation of local growth factors. Genetic factors are highly important, defining host susceptibility through polymorphisms, for example, related to the enzymes that affect hormone levels, estrogen/progesterone receptors, and protein synthesis [18]. DNA methylation can mimic the effects of germline mutations in cancer predisposition genes such as breast cancer type 2 susceptibility (BRCA2) [19]. A variety of interrelated genetic, environmental, and physiological factors appears to be associated with increased risk of breast cancer, but no single factor or combination of variables presently known is sufficient to explain the etiology of the disease [20]. While androgens have been described as mutagens, estrogens could become endogenous carcinogens via the formation of catechol estrogen quinones, which react with DNA to form specific depurination estrogen-DNA adducts [21], possibly inducing cell transformation and initiation of BCa [21]. Current knowledge about the most aggressive forms of BCa points out the role of specific cells with stem properties located within the tumor and called BCa stem cells [22]. Interestingly, sex steroid receptors involved in BCa etiology and progression could promote BCa stem cell proliferation, dedifferentiation, and migration [22], even though the molecular mechanisms allowing this have not been fully deciphered so far.

Local therapy surgery remains the first step in the treatment of BCa; besides, the development of new conservative procedures has improved the patients’ quality of life as well as their life expectancy [23]. Thereafter, neoadjuvant chemotherapy or adjuvant therapy depends on the tumor characteristics such as cell growth rate, tumor grade, or lymph nodes dissemination. Based on the fact that estrogens have an important role in luminal cell growth, endocrine therapy has been historically developed at the beginning of the 70′s for blocking estrogen receptors with the use of SERMS or SERDs (for an extensive review see [24]). As for PCa, the appearance of a metastatic stage of the disease drastically decreases the life expectancy of the patients. In that situation and together with increasing survival, the aim of the therapy is to maintain the quality of life and to reduce the symptoms. Accordingly, “international guidelines” recommend endocrine therapy as the first therapeutic choice in patients with human epidermal growth factor receptor 2 (HER2)-negative luminal metastatic BCa unless a visceral crisis or another life-threatening situation requires chemotherapy [25]. These treatments, targeting the estrogen signaling pathways include SERMs, SERDs, and aromatase inhibitors [24].

## 3. The Activity of Androgens and Estrogens

Altogether, 5α-reductase, aromatase, and 17βHSD are theoretically the main enzymes to be targeted to decrease the levels of DHT and 17βE2 and to reduce the progression of hormone-dependent PCa and BCa.

Once synthesized, DHT and 17βE2 act through their cognate nuclear receptors AR (nuclear receptor subfamily 3, group C, member 4, NR3C4) and ERα (NR3A1), respectively. ERβ (NR3A2) is a second estrogen receptor; however, conversely to ERα whose expression increases at the early stages of BCa and acts as a tumor promoter, ERβ is reduced during carcinogenesis and cancer progression and seems to act as a tumor suppressor [26]. Altogether, these nuclear receptors display ligand-modulated transcription [27]. The binding of DHT or 17βE2 within the ligand-binding pocket of AR or ERα, respectively, induces a conformational modification of the receptor, it’s shuttling from the cytoplasm to the nucleus, and its binding to specific DNA sequences located in its target genes. This binding and the recruitment of co-activators regulate the expression of specific genes involved in breast and prostate epithelium homeostasis [28]. Hence, Nelson et al. identified a program of androgen-responsive genes in the neoplastic prostate epithelium [29] and classified them into several physiological pathways: (i) metabolism, such as those regulating the fatty acid (sterol response element-binding protein, fatty acid synthase, stearoyl-CoA desaturase) and the cholesterol (3-hydroxy-3-methyl-glutaryl-coenzyme A (HMG-CoA)-synthase, HMG-CoA-reductase, 3-β-hydroxysterol-Δ-24 reductase) homeostasis; (ii) transport or trafficking, such as the transcript encoding the FK-506 binding-protein FKBP5 (FKBP51); (iii) cell proliferation or differentiation. AR is not the only nuclear receptor that could be involved in the progression of PCa. Hence, Gail Prins’ group has shown that early exposure to estrogens and estrogen-like compounds could also increase PCa incidence through ERα [30,31].

A parallel can again be drawn between PCa/androgens and BCa/estrogens. Approximately 70% of all BCa express ERα, progesterone receptor (PgR; NR3C3), or both, and such tumors are considered hormone receptor-positive [32]. Several genetic programs have linked estrogens to BCa (for a review, see [33]). For example, 17βE2 facilitates the G1/S phase transition by the over-accumulation of cyclin D1 and its activation or could also modulate the tumor surrounding immune cells. Not only 17βE2 can drive ERα activity; indeed, Siersbaek et al. [34] pointed out that membrane-receptors and other steroid receptors could also modulate ERα function in BCa.

Targeting androgen synthesis and the AR pathway has been and remains central to PCa pharmacology therapy [35,36]. 5α-androstane-3β, 17β-diol (3β-Adiol), synthesized from testosterone in the prostate, inhibit PCa cell proliferation, migration, and invasion, acting as an anti-proliferative/anti-metastatic agent. Hence 3β-Adiol is unable to bind AR; it exerts its protective activity by interacting with ERβ [37]. Interestingly, the combination of the phytoestrogens genistein, quercetin, and biochanin A inhibits the growth of androgen-responsive prostate cancer cells (LNCaP) as well as DU-145 and PC-3, two androgen-insensitive prostate cancer cells [38]. Subsequent mechanistic studies in PC-3 cells indicated that the action of phytoestrogens was mediated both through ER-dependent and ER-independent pathways [38]. Bicalutamide that acts as a pure antagonist in parental LNCaP cells showed agonistic effects on AR transactivation activity in LNCaP-abl cells and was not able to block the effects of androgen in these cells. The non-steroidal AR blocker hydroxyflutamide exerted stimulatory effects on AR activity in both LNCaP and LNCaP-abl cells; however, the induction of reporter gene activity by hydroxyflutamide was 2.4- to 4-fold higher in the LNCaP-abl subline [39].

Lead compound 16-(4,6-Dimethyl-1,2-dihydro-1,3,5-triazin-2-yl)-17-chloro-Δ1,3,5(10), 16-estratetraen-3-ol displayed selectivity in ERα-positive breast cancer cells. At 10 μM concentration, this heterosteroid inhibited 50% of the E2-mediated ERα activity and led to partial ERα down-regulation. Docking studies suggested that the binding mode of this molecule was within the ER pocket [40].

## 4. Pharmacological Treatments of BCa and PCa

Altogether and based on the previous data, the main pharmacological treatments aim to reduce the levels of active steroids by (i) inhibiting in vivo synthesis from the cholesterol, (ii) blocking AR and ERα by specific antagonists and (iii) specific degraders (Figure 2). Note that the inhibition of de novo cholesterol synthesis by statins will not be covered in this review (Table 1).

### 4.1. Modulation of the Enzymes Involved in 17βE2 and DHT Synthesis

Inhibition of aromatase, the enzyme responsible for converting androgens to estrogens, is therapeutically useful for the endocrine treatment of hormone-dependent BCa [41] or PCa [42]. Melatonin, at physiological concentrations, decreases aromatase activity and expression in the human breast cancer cells MCF-7 [43]. A cell-free in vitro assay confirmed that melatonin, as well as 2-methyl indole hydrazones, binds the catalytic site of aromatase [44]. The synthetic aromatase inhibitor and steroid-derivative exemestane [45] is prescribed to postmenopausal women with advanced BCa whose disease has progressed despite tamoxifen therapy [46]. Anastrozole [47], is also a potent aromatase inhibitor. Even though it does not use the same mechanistic as exemestane, anastrozole displays the same effect as adjuvant treatment for hormone receptor-positive early BCa [48]. New aromatase inhibitors based on the testosterone skeleton could decrease aromatase stability, disrupt the cell cycle progression of MCF7 cells, and induce their apoptosis through the mitochondrial pathway [49]. The same efficacy and effects were found with 7α-substituted steroid molecules [50]. In addition, the steroids 3β-hydroxyandrost-4-en-17-one, androst-4-en-17-one, 4α,5α-epoxyandrostan-17-one, and 5α-androst-2-en-17-one, obtained from modifications in the A-ring of androstenedione, inhibit aromatase, and decrease cell viability [51]. Altogether, 5α-androst-3-en-17-one and 3α, 4α-epoxy-5α-androstan-17-one are the most potent irreversible aromatase inhibitors [52].

Even though currently recommended in clinical guidelines for benign prostatic hyperplasia or for the treatment of androgenic alopecia, the potential use of 5αR inhibitors has been questioned these last years in PCa. Indeed, as indicated in Figure 2, the possible intratumoral reduction of testosterone into the potent DHT, which critically contributes to the progression of PCa and its castration-resistant stage, has driven this hypothesis. Finasteride [53] and its analog dutasteride [54] are the two 4-azasteroids that decrease the prostatic DHT concentration by 85 to 90%. Finasteride has been described to lower the risk of low-grade PCa but seems to increase the risk of high grade, and has no effect on overall survival. Based on a limited number of patients, dutasteride could be more efficient in treating CRPC [55]. However, altogether, no impact of 5αR inhibitor on survival has been found in people with PCa [56].

17β-hydroxysteroid dehydrogenases (17βHSDs) catalyze the interconversions between active 17β-hydroxysteroids and less-active 17-ketosteroids and modulate the availability of biologically active estrogens and androgens in breast and prostate [57]. Among them, 17βHSD type 1 is essential for the production of 17βE2. These enzymes are thus theoretically exquisite targets to reduce the production of 17βE2. However, if several steroidal and non-steroidal compounds are able to reduce HSD17B1 activity in vitro, the list of in vivo validated inhibitors is much shorter [58], and finally, there is no 17βHSD type 1 inhibitor currently used for the treatment of BCa.

17α-hydroxylase/C17-20-lyase (CYP17A1) converts pregnane steroids into androgens like testerosterone. Abiraterone acetate [59] is used in metastatic castration-resistant PCa [60] to block the biosynthesis of androgens by inhibiting CYP17A1 activity.

### 4.2. Antagonists of ERα and AR Transcriptional Activities

The use of selective estrogen receptor modulators (SERMs) dates to the late 1960s and early 1970s when positive clinical outcomes were reported with the use of antiestrogenic agents such as tamoxifen [61], a *trans*-isomere of 1(p-β-dimethylaminoethoxy-phenyl)-1-2-diphenylnut-1-ene, which has complex pharmacology. Apart from being metabolized into numerous biologically active compounds, it is an estrogen agonist-antagonist depending on its competitive binding to ERα. Raloxifene, a benzothiophene SERM, initially failed to be used in women treated with a BCa; however, it is now used to decrease the incidence of invasive BCa in postmenopausal women who have a higher risk to develop the disease [62]. SERMs tamoxifen and raloxifene were also approved for the chemoprevention in women with a high risk of breast cancer [63].

Based on the analogy with SERMs, SARMs have been developed to block the transcriptional activity of AR. Non-steroidal antiandrogens were first introduced in 1989 in clinical practice as a treatment for advanced and metastatic PCa [64]. The first-generation of antiandrogens bicalutamide [65], flutamide [66], and nilutamide [67] efficiently block AR, and thereby, inhibit tumor growth with similar efficacy, even though bicalutamide is better tolerated and more stable antiandrogen currently used in clinical practice [68]. The second generation of SARMs is represented by enzalutamide [69]. In contrast to the first generation of AR blockers, enzalutamide also inhibits the shuttling of AR from the cytoplasm to the nucleus, and thus impairs AR binding to DNA [70]. Since enzalutamide, apalutamide and darolutamide [71] have been approved by FDA and EMA for the treatment of metastatic PCa and non CRPC, respectively.

### 4.3. Antagonists of the Steroid Receptors and Inducers of Their Degradation

While ERα and AR blockers have been currently used for decades, it came to the evidence that these nuclear receptors were able to go back to a transcriptionally active state despite the presence of their specific antagonists, hence making the tumor resistant to chemical castration. The idea was, thus, to develop a molecule able to inhibit the binding of the bona fide ligand to the ligand-binding pocket of the receptor and, at the same time, to induce its proteasomal degradation. Hence, the possibility for the remaining receptor to be activated even in the presence of the antagonist would have been decreased.

Fulvestrant, a 7α-alkylsulphinyl analog of 17βE2, is distinctly different in chemical structure from the nonsteroidal structures of tamoxifen, raloxifene, and other SERMs. The binding affinity of fulvestrant to ERα is 89% that of the natural ligand [72] and significantly greater than the affinity of tamoxifen. SERD of first-generation, fulvestrant, also impairs ER-dimerization and its translocation to the nucleus. More importantly, the unstable fulvestrant-ERα complex is unstable and rapidly degraded, inducing in parallel a decrease in the amount of ERα encoding mRNA [73], which explains its robustly effective antitumor activity in preclinical models of BCa [74]. Fulvestrant was first approved in 2002 as monotherapy for the treatment of postmenopausal patients with positive ER metastatic BCa whose cancer had progressed following anti-estrogen therapy [75]. Furthermore, neoangiogenesis is impaired by intraductal fulvestrant treatment [76]. When added to anastrozole, fulvestrant co-treatment improves the overall survival of patients with metastatic hormone receptor-positive BCa compared with anastrozole alone [77]. Novel SERDs have also been designed based on the 6-OH-benzothiophene scaffold common to arzoxifene, another SERM, and raloxifene [78]; however, these molecules did not reach clinical use so far.

Conversely to the clinically available SERDs, no equivalence of specific androgen receptor degraders or down-regulators (SARDs) have been used so far in patients with a PCa. However, darulotamide derivatives have been screened in cell culture. It appears that two isolated compounds have the ability to act as specific AR antagonists more efficiently than enzalutamide, as well as down-regulators in inducing the degradation of the receptor and decreasing AR mRNA [79]. These quinoline or purine derived-darulotamide could, thus, be considered as lead compounds for the synthesis of new SARDs to be used for the treatment of CRPC.

## 5. Natural Compounds Modulating the Steroid Activity

Several steps could be targeted (Figure 2) to decrease or to abolish androgens and estrogens activity on their respective nuclear receptors in PCa or in BCa. It is impossible to list all the molecules used in folk medicine and suspected to modify the activity of the steroids. However, some significant examples have been chosen from the published literature (Table 1).

### 5.1. Natural Compounds Inhibiting the Steroid Synthesis

Fifteen natural products were screened in traditional Chinese medicine [80], and seven of them showed potent inhibition of aromatase: naringin, apigenin, berberine, palmatine, bavachin, jatrorrhizine and bavachinin. These compounds have been classified as flavone (apigenin), flavonone (naringin, bavachin, bavachinin), or isoquinoline alkaloid (berberine, palmatine, jatrorrhizine). On these bases, indole derivatives have been synthesized as aromatase inhibitors [81]. *Alangium salviifolium*, commonly known as sage-leaved alangium, is a flowering plant in the Cornaceae family. Most of the isolated cardinane sesquiterpenes are potent aromatase inhibitors [82]. The modulation of the flavonoid skeleton increases the anti-aromatase effect [83]. The hexane extract of the leaves of *Brassaiopsis glomerulata*, a large shrub found in Indonesia, presents significant aromatase inhibition [84]. *Psoralea corylifolia*, a plant used in Indian and Chinese traditional medicines, contains a significant amount of bakuchiol, a meroterpene used for its antiandrogenic activity, which shows moderate anti-aromatase activity [85]. *Sarcococca saligna*, the sweet box or Christmas box, native from Northern Pakistan, contains phytochemical constituents such as alkaloid-C, dictyophlebine, sarcovagine-D and saracodine, which inhibit aromatase [86]. Shu-Gan-Liang-Xue decoction (SGLXD) is a traditional Chinese herbal formula with a potential dual aromatase-sulfatase inhibitor by simultaneously down-regulating the expressions of aromatase and sulfatase in BCa cell cultures [87]. SGLXD also has anti-tumor effects on the BCa cells ZR-75-1 by inhibiting aromatase and steroid sulfatase [88]. Butein, a natural chalcone found *in Toxicodendron vernicifluum*, shows anti-aromatase activity and could potentially represent a natural alternative for the chemoprevention or therapy of BCa [89]. Not only green plants but also white button mushroom could contain interesting phytochemicals as their aqueous extract inhibits aromatase activity and proliferation of MCF-7 cells [90]. Obacunone, a natural limonoid present in citrus fruits, affects aromatase activity, increases apoptosis, and induces G1 cell cycle arrest [91].

Even though the use of 5αR inhibitors in PCa is controversial, it is noteworthy that numerous extracts present the capacity to inhibit the bioconversion of testosterone into DHT. Historically, lipophilic extracts of *Serenoa repens*, commonly known as saw palmetto, were the first to be associated with a strong inhibition of the DHT synthesis [92]. *Epilobium angustifolium*, a native of the temperate northern hemisphere, could present some 5α-reductase inhibition as its ethanolic extract directly inhibits the proliferation of the PZ-HPV-7 cells that are sensitive to DHT [93]. Hiipakka et al. [94] identified the green tea (−)-epigallocatechin gallate as well as other natural flavonoids such as biochanin A, daidzein, genistein and kaempferol as potent inhibitors of the 5α-reductase type 2. *Adina rubella*, a shrub found in Korea, presents caffeic acid and grandifloroside from leaves that have potent inhibitory activity against 5α-reductase [95]. *Cynomorium songaricum*, a parasitic perennial flowering plant, contains polyphenolic constituents that significantly inhibit rat prostate enlargement, improved the pathological feature and reduced the thickness of the smooth muscle layer [96]. The mechanism encompasses, in part, the inhibition of the 5αR activity, together with the decrease of AR and ERα mRNA accumulation and the increase of ERβ mRNA [96]. Altogether, and despite the until now low potential of using 5αR inhibitors in the treatment of PCa, the discovery of specific inhibitors from natural extracts is still of interest [97].

### 5.2. Natural Compounds Acting as AR or ERα Antagonists

So far, only a few natural compounds have been described as having an anti-AR activity. Methanolic extracts of the bark of *Brosimum rubescens*, also known as *palo de sangre* in Peru, has shown potent inhibitory activity towards DHT binding to AR [98]. Tanshinone IIA, one of the most abundant constituents of the root of the Chinese sage *Salvia militiorrhiza*, reduces the accumulation of ERα and AR mRNA in prostate cell lines. In fact, Tanshinone IIA can inhibit the growth of stromal and epithelial cells in vitro and in vivo by a mechanism that may involve arresting the cell cycle and downregulating ERα and AR accumulation [100]. The fact that AR is downregulated is in favor of a SARD role for Tanshinone IIA. We studied the effects of the ethanolic extract of propolis on the phenotype of LNCaP cells [101]. This extract reduces cell survival, induces apoptosis, and blocks the cell cycle at the G0/G1 phase. Interestingly, this ethanolic extract decreases the accumulation of AR and the secreted amount of the androgen target prostate-specific antigen (PSA) in LNCaP cells. This anti-androgen activity was also shown on the androgen target genes *Fkbp5* and *Sgk1*. Finally, the capacity of propolis to block AR functioning was demonstrated in transient transfections using the human AR. Altogether, the ethanolic extract of propolis displays SARD activity that needs to be further investigated in preclinical models [101]. The 3,3′-diindolylmethane is a major digestive product of indole-3-carbinol, a potential anticancer component of cruciferous vegetables. It suppresses cell proliferation of LNCaP cells and inhibits the stimulatory effect of DHT on DNA synthesis. It also blocks AR translocation into the nucleus by strong competitive inhibition of the DHT binding [99]. Bisbibenzyl compounds riccardin C and D as well as marchantin, all extracted from various liverwort species, decrease AR expression at mRNA and protein levels, leading to the suppression of AR transcriptional activity. However, these effects seem to be linked to proteasome inhibition and autophagy activation in LNCaP cells rather than to the AR-binding effect [104].

In BCa, a cycloartane triterpenoid isolated from *Schisandra glaucescens*, a magnolia vine native to Asia and North America, shows ER antagonistic effects [102]. Otherwise, phytoestrogens, mycoestrogens, and xenoestrogens bind ER in intact cells, but display marked differences in their ability to induce end products of estrogen action and to regulate cell proliferation [105]. All of the different classes of these estrogen-like molecules stimulate cell proliferation at concentrations that half-saturated ER; the fact that EC_50_s are lower than those of 17βE2 explain their slight antagonist effects in the presence of the natural ligand, as exemplified by genistein found in soy foods [105]. Soy phytoestrogens are non-steroid molecules whose structural similarity lends them the ability to mimic with a lower efficacy the effects of 17βE2 [106]. Extracts from *Urtica dioica*, often known as common nettle, has some anti-estrogen activity [103]; however, its active compound has not been identified so far. Altogether, natural estrogen-like molecules are numerous. To help in the screening of which ones could have a pro- or an anti-ER activity, Powers and Setzer [107] have developed a molecular docking approach to identify the estrogen activity of phytochemicals, which allowed the study of almost 600 of them. They identified that the prenylation of flavonoids often results in anti-estrogenic activity.

## 6. Conclusions

Altogether, molecules extracted from a natural environment are the source of the identification of compounds that could serve as lead compounds for building more active drugs. Interestingly, such natural compounds can target the various levels that control androgen or estrogen concentration, AR and ERα transcriptional activity and degradation. The main negative point is, however, that the great majority of these compounds have been tested in vitro or in cell culture, but few have been studied in preclinical models. On the other hand, it is difficult from an ethical point of view to study thousands of these molecules in preclinical models. One solution would be to perform a pre-screening using in silico models and to realize in vivo tests in an ex vivo 3D model [108] or in non-mammalian models of tumors, such as the drosophila for PCa [109].

## Figures and Tables

**Figure 1 ijms-21-03690-f001:**
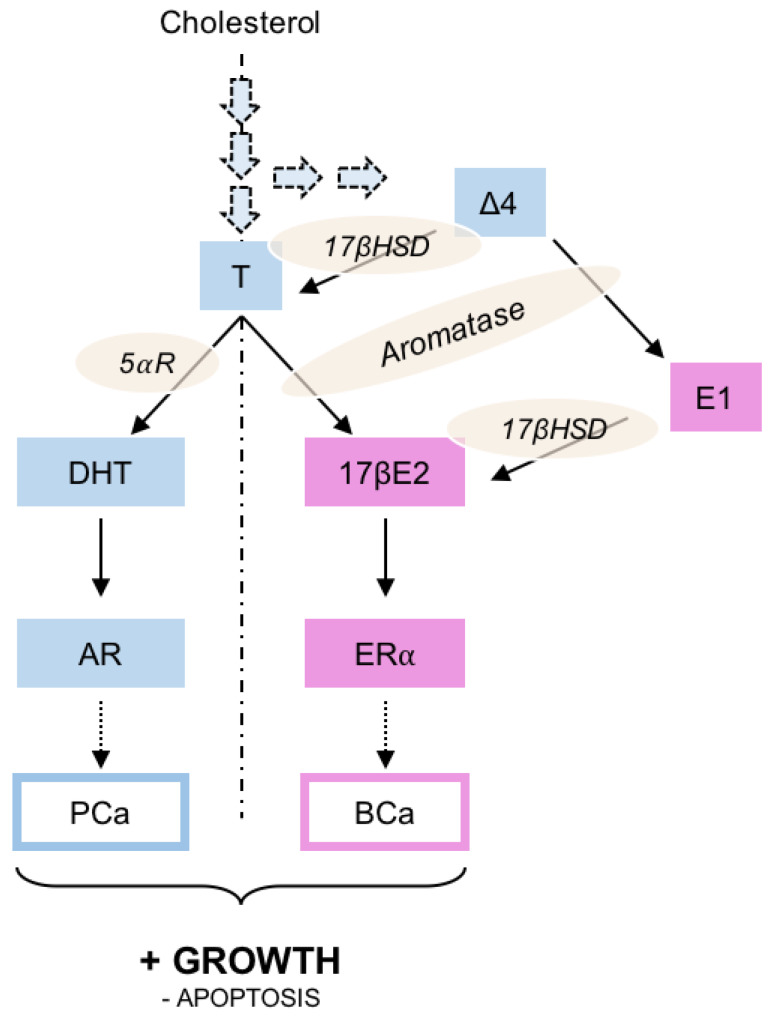
Schematic representation of the steroid activity on prostate (PCa) or breast (BCa) cancer. Three various levels of control could be pointed out: i) steroid synthesis by inhibiting the enzymatic pathways leading to testosterone or 17β-estradiol production; ii) modulation of the androgen or estrogen receptor activity using specific antagonists; iii) modulation of the steroid receptor activity together with the induction of its degradation/down-regulation. Δ4, Δ4-androstenedion; 5⍺R, 5⍺-reductase; 17βE2, estradiol; 17βHSD, 17β-hydroxy-steroid dehydrogenase; AR, androgen receptor; BCa, breast cancer; E1, estrone; ER⍺, estrogen receptor ⍺; PCa, prostate cancer; T, testosterone.

**Figure 2 ijms-21-03690-f002:**
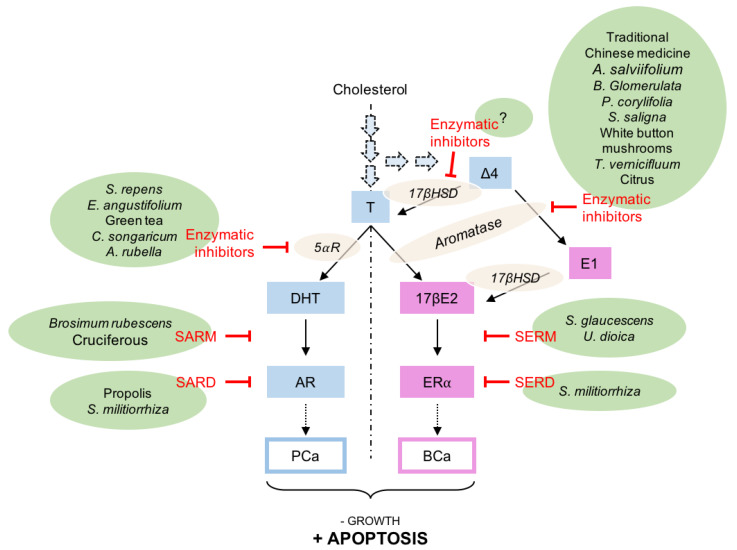
Examples and sites of activity of natural extracts that could modulate androgen and estrogen receptor activities in prostate (PCa) or breast (BCa) cancer. For more information regarding the isolated molecules that show a significant activity, refer to the main text. Δ4, Δ4-androstenedione; 5⍺R, 5⍺-reductase; 17βE2, estradiol; 17βHSD, 17β-hydroxysteroid dehydrogenase; AR, androgen receptor; BCa, breast cancer; E1, estrone; ER⍺, estrogen receptor ⍺; PCa, prostate cancer; SARM, specific androgen receptor modulator; SARD, specific androgen receptor degrader/down-regulator; SERM, specific estrogen receptor modulator; SERD, specific estrogen receptor degrader/down-regulator; T, testosterone.

**Table 1 ijms-21-03690-t001:** Examples of molecules for the synthesis and activity of androgens and estrogens.

Pharmacological Targets	Synthetic Compounds	Natural Compounds
Aromatase	Exemestane [45] Anastrozole [47] 7α-substituted steroids [50]. Δ4-androstenedione derivatives [51].	Melatonin [44] Naringin, apigenin, berberine, palmatine, bavachin, jatrorrhizine, bavachinin [80] Alangenes [82] Extracts of *Brassaiopsis glomerulata* [84] Bakuchiol [85] Extracts of *Sarcococca saligna* [86] Shu-Gan-Liang-Xue decoction [87,88] Aqueous extracts of white button mushrooms [90] Butein [89] Obacunone [91]
5α-reductase	Finasteride [53] Dutasteride [54]	*Serenoa repens* extracts [92] Ethanolic extracts of *Epilobium angustifolium* [93] (−)-epigallocatechin gallate, biochanin A, daidzein, genistein, kaempferol [94] caffeic acid, grandifloroside [95] polyphenols from *Cynomorium songaricum* [96]
Androgen receptor (SARM)	Bicalutamide [65] Enzalutamide [69] Darolutamide [71]	Methanolic extract of *Brosimum rubescens* bark [98] 3,3′-diindolylmethane [99]
Androgen receptor (SARD)	Darolutamide derivatives [79]	Tanshinone IIA [100] Ethanolic extracts from propolis [101]
Estrogen receptor (SERM)	Tamoxifen [61] Raloxifene [62]	Extracts of *Schisandra glaucescens* [102] Extracts of *Urtica dioica* [103]
Estrogen receptor (SERD)	Fulvestrant [72] 6-OH benzothiophene [78]	Tanshinone IIA [100]

SARD, specific androgen receptor degrader/down-regulator; SARM, specific estrogen receptor modulator; SERD, specific estrogen receptor degrader/down-regulator; SERM, specific estrogen receptor modulator. For more information, refer to the main text.

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
