# Peer review of "Pharmacological Modulation of Steroid Activity in Hormone-Dependent Breast and Prostate Cancers: Effect of Some Plant Extract Derivatives"

_ijms, 2020, doi:10.3390/ijms21103690_

Round 1

Reviewer 1 Report

The manuscript by Bayala et al titled “Modulation of steroid activity in hormone-dependent breast and prostate cancers by plant extract derivatives” is a review of current literature. Overall it is well researched and contains relevant and significant information. This reviewer has only a couple of minor issues.

There are many overly grand statements throughout the manuscript that should be edited. For example, the very first line of the abstract “Breast and prostate tumors are paradigmal hormone dependent cancers.” Many are, but not all breast cancers are hormone-dependent. Such statements appear throughout. HER2+ and triple-negative breast cancers are not considered hormone-dependent and these cases make up approximately 15% of all breast cancer diagnoses.

The second issue is the English grammar and sentence structure. The manuscript requires significant editing. The information is conveyed but in a manner that makes it difficult to understand initially.

Author Response

Reviewer #1. Modifications in the manuscript have been enlightened in green.

The manuscript by Bayala et al titled “Modulation of steroid activity in hormone-dependent breast and prostate cancers by plant extract derivatives” is a review of current literature. Overall it is well researched and contains relevant and significant information. This reviewer has only a couple of minor issues.

** We would like to thank Reviewer #1 for the positive comments regarding this review.

There are many overly grand statements throughout the manuscript that should be edited. For example, the very first line of the abstract “Breast and prostate tumors are paradigmal hormone dependent cancers.” Many are, but not all breast cancers are hormone-dependent. Such statements appear throughout. HER2+ and triple-negative breast cancers are not considered hormone-dependent and these cases make up approximately 15% of all breast cancer diagnoses.

** We agree with Reviewer #1 that this sentence is probably too exaggerated. We have modified it in order to incorporate Reviewer #1’s comment. Furthermore, we have also amended assertions suggesting that all PCa and BCa are hormone-dependent tumors. We hope that the new version will be more reflecting the reality of BCa and PCa.

The second issue is the English grammar and sentence structure. The manuscript requires significant editing. The information is conveyed but in a manner that makes it difficult to understand initially.

** We apologize for this unacceptable fact. According Reviewer #1’s comment, this article has been edited by a fluent native English speaker (modifications in purple). We hope that this review will be easier to read now.

Besides, note that we have deleted the beginning of chapter 3, as it was redundant with what was described in the introduction.

Reviewer 2 Report

Based on the title of the manuscript and the type of journal, the reader would expect a complete overview on plant extracts which are able to modulate steroid activity, with a focus on their molecular mechanisms. However, the discussion of plant extracts is limited to a minimally commented list in the last section of the manuscript. The authors could at least focus a little bit more on the most promising candidates, based on the level of available evidence, the safe and effective use in other pathologies, the knowledge of additional mechanisms of action which could support an antiproliferative activity, etc. In its present form, the title is misleading.

Figure 1 shows hormones’ production pathways, the title of the figure is focused on steroid activity, while the legend is focused possible mechanisms of control. SARM, SARD, SERM and SERD are cited in the legend but are not shown in the figure. There should be more consistency between the figure, its title and the legend. Maybe figure 2 could be sufficient.

A substantial revision of English language is needed. There are several misspellings, grammar and syntax mistakes.

Author Response

Reviewer #2. Modifications in the manuscript have been enlightened in yellow.

Based on the title of the manuscript and the type of journal, the reader would expect a complete overview on plant extracts which are able to modulate steroid activity, with a focus on their molecular mechanisms. However, the discussion of plant extracts is limited to a minimally commented list in the last section of the manuscript. The authors could at least focus a little bit more on the most promising candidates, based on the level of available evidence, the safe and effective use in other pathologies, the knowledge of additional mechanisms of action which could support an antiproliferative activity, etc. In its present form, the title is misleading.

** we agree with Reviewer #2’s comments that, as it, the title is misleading. Accordingly, we have amended it. Likewise, we have also modified the abstract and the end of the introduction, specifying that we focused only on some promising candidates. We had previously added within the manuscript that the presented plant extracts were some examples among many others (cf. beginning of chapter 5) as it is quite impossible to list all the extracts without being boring.

Figure 1 shows hormones’ production pathways, the title of the figure is focused on steroid activity, while the legend is focused possible mechanisms of control. SARM, SARD, SERM and SERD are cited in the legend but are not shown in the figure. There should be more consistency between the figure, its title and the legend. Maybe figure 2 could be sufficient.

** We apologize for the mistake within the legend of Figure 1. It has been corrected accordingly. We however still believe that 2 figures are necessary. As Reviewer #2, we thought to add only one figure at the beginning. However, the reading of the manuscript was more difficult due to the distance in the main text. After testing both solutions, it came up that 2 figures were a better option.

A substantial revision of English language is needed. There are several misspellings, grammar and syntax mistakes.

** We apologize for this unacceptable fact. According Reviewer #2’s comment, this article has been edited by a fluent native English speaker (modifications in purple). We hope that this review will be easier to read now.

Besides, note that we have deleted the beginning of chapter 3, as it was redundant with what was described in the introduction.

Round 2

Reviewer 2 Report

The authors modified the title and the aim of the review as suggested. It is a pity they did not try to improve the description of candidate natural compounds.

English language has been significantly improved; a check for minor leftover errors is suggested.